# Targeted transcriptome analysis using synthetic long read sequencing uncovers isoform reprograming in the progression of colon cancer

Silvia Liu[1,2,3,5], Indira Wu[4,5], Yan-Ping Yu [1,2,3], Michael Balamotis[4], Baoguo Ren[1,2], Tuval Ben Yehezkel[4✉] & Jian-Hua Luo [1,2,3✉]

The characterization of human gene expression is limited by short read lengths, high error rates and large input requirements. Here, we used a synthetic long read (SLR) sequencing approach, LoopSeq, to generate accurate sequencing reads that span full length transcripts using standard short read data. LoopSeq identified isoforms from control samples with 99.4% accuracy and a 0.01% per-base error rate, exceeding the accuracy reported for other long-read technologies. Applied to targeted transcriptome sequencing from colon cancers and their metastatic counterparts, LoopSeq revealed large scale isoform redistributions from benign colon mucosa to primary colon cancer and metastatic cancer and identified several previously unknown fusion isoforms. Strikingly, single nucleotide variants (SNVs) occurred dominantly in specific isoforms and some SNVs underwent isoform switching in cancer progression. The ability to use short reads to generate accurate long-read data as the raw unit of information holds promise as a widely accessible approach in transcriptome sequencing.

[1] Department of Pathology, University of Pittsburgh School of Medicine, Pittsburgh, PA 15261, USA. [2] High Throughput Genome Center, University of Pittsburgh School of Medicine, Pittsburgh, PA 15261, USA. [3] Pittsburgh Liver Research Center, University of Pittsburgh School of Medicine, Pittsburgh, PA 15261, USA. [4] Loop Genomics, Inc., San Jose, CA 95138, USA. [5] These authors contributed equally: Silvia Liu, Indira Wu. ✉email: tuval@loopgenomics.com; luoj@upmc.edu

The development of massively parallel short-read sequencing and their overlapping alignment in the last 20 years has made it possible to decipher the genome sequences of numerous organisms[1–3]. It has also enhanced the quantification of gene expression by allowing the sequencing of millions of transcripts at an affordable price point. For mammalian cells, the transcription of a gene involves an alternative splicing process that selectively utilizes specific exons while removing other exons and introns from the final transcript[4]. This generates many different transcripts (isoforms) with altered amino acid sequences from the same gene and dramatically increases the diversity of the gene products[5]. However, given the high sequence homology between the different isoforms from the same gene, the characterization and quantification of isoforms using short-read sequencing is challenging as the span of the short-reads typically renders isoform mapping and identification ambiguous. Similarly, determining the exact exon composition of a single mRNA molecule is difficult without longer sequencing lengths or the ability to trace the short-reads to the originating individual transcripts.

To overcome the read-length limitation of short-read sequencers, various approaches of synthetic long-read (SLR) sequencing have been developed and applied to various difficult-to-sequence applications, including transcriptome sequencing[6]. SLR sequencing methods rely on binning short-reads based on barcoding information assigned to individual transcripts during library preparation and SLRs are reconstructed from clusters of short-reads that share the same barcode. One SLR approach involves physically partitioning DNA molecules into multi-well plates or microfluidic droplets, typically in the order of hundreds of molecules per partition, with the DNA molecules within each partition assigned the same barcode[7–9]. While this approach demonstrates the power of SLR by grouping short-reads from molecules of the same well/droplet into one de novo assembly, the process of physically partitioning molecules is labor-intensive and cannot resolve the sequences of homologous transcripts (such as isoforms) within the same well/droplet. Here we present the application of SLRs to isoform sequencing using LoopSeq, a SLR technology that leverages Illumina short-read sequencing platforms and resolves some of the major drawbacks of previously developed SLR approaches[7–11] to generate accurate long-reads from mRNA.

We validated the accuracy of LoopSeq by sequencing known isoforms in control samples, compared it to the results from sequencing the same control samples from other long-read and SLR technologies, and demonstrated its utility in discovering fusion gene isoforms, in quantifying isoform distributions, and in discovering mutation isoform expression patterns in sets of clinical samples.

## Results

### The strategy of LoopSeq.
LoopSeq employs unique molecular identifiers (UMIs) instead of well/droplet identifiers, which are randomly and intramolecularly distributed along the length of barcoded molecules. As shown in Fig. 1A, it first assigns an UMI to each first-strand cDNA molecule during reverse transcription. Following this barcoding step, a probe-capture-based target enrichment step is applied to select for cDNA molecules of interest. Following capture and PCR amplification of the barcoded cDNA, UMIs are randomly transposed to various internal positions of the molecules, and the sequence immediately adjacent to the UMI insertion site is converted into an Illumina short-read that contains both the UMI and the adjacent sequence. After short-read sequencing, short-reads tagged with identical sample indices and UMI's are binned and used for de novo SLR assembly

which generates a single long-read for each barcoded cDNA molecule. Specifically, the short-read libraries are sequenced using PE150 Illumina chemistry, trimmed using Trimmomatic (see 'Methods'), then binned by sample index and by UMI. Each cluster of short-reads that share the same sample index and UMI is de novo assembled into a long-read using SPADES[12] (see Supplementary script). This enables the reconstruction of single, long contiguous molecules from short sequencing reads, even in samples that contain mixtures of highly homologous long-molecules such as mRNA isoforms or RNA editing variants. Additionally, it does not require the physical partitioning of RNA molecules into different wells or microfluidic droplets. It improves on previous UMI-based SLR approaches[10,11] by (a) reducing the error rates due to more uniform short-read coverage of each transcript (see 'LoopSeq ERCC error rates' section below), (b) reporting SLR data quality in FASTQ format with per nucleotide Q scores (see section 'LoopSeq ERCC error rates'), and (c) introducing known synthetic terminal 5′ and 3′ adapters that both enable the distinction between partially reconstructed SLRs and fully reconstructed SLRS as well as improve the quality of 3′ and 5′ transcript de novo assembly. A SLR is categorized as a full-length read if both the 5′ and 3′ synthetic adapters are identified at the termini of the SLR. If a molecule is not full length, the adapters are used to report whether the SLR is missing either 5′ or 3′, or both. Termini detection in SLR transcriptome data is critical for the ability to differentiate between partially reconstructed SLRs and true previously unknown transcription start site (TSS) and transcription termination site (TTS) found in fully reconstructed SLRs. Additionally, modification of the transcript sequence with known synthetic sequences at both termini trivializes (1) the identification of the true terminal sequences by looking for the sequences immediately adjacent to the adapters and (2) determining whether the SLR is full length since full-length SLRs should have the adapters present at the termini.

### ERCC transcript completeness.
To demonstrate that LoopSeq SLRs cover the full-length of cDNA molecules, we prepared and sequenced 7481 synthetic RNA control ERCC cDNA molecules alongside 27,426 mRNA molecules from Hela total RNA. After SLR reconstruction for each uniquely tagged cDNA molecule, the TSS and the TTS of each ERCC transcript identified as full-length were compared to the reference sequences. Histograms of the TSS and TTS differences are shown in Fig. 1B, C. For TSS, 82.6% of the full-length contigs correctly identified the start site of the cDNA, and 12.6% of the full-length contigs report a TSS that is downstream of the annotated TSS, based on the designation of a full-length contig, that is the reconstructed sequence reaches the expected adapter sequence at both the 5′ and the 3′ ends of the cDNA molecule. If a cDNA molecule is prematurely terminated during reverse transcription, either due to degraded RNA or non-specific terminal transferase activity of the reverse transcriptase, the reconstructed molecule would include the 5′ adapter sequence, correctly identifying that the full-length cDNA molecule is reconstructed, while having a TSS that is downstream of the annotated TSS. The LoopSeq method uses a template-switching oligo to capture the TSS, which in practice has the ability to attach to pre-terminated molecules, giving them the false appearance of being full length. Nearly all non-canonical TSSs identified were shifted greater than 100 bp downstream, consistent with premature termination. For the TTS, 68.7% of the full-length contigs correctly identified the termination site of the cDNA, while 15.6% of the full-length contigs had a TTS that was within 5 nucleotides from the annotated TTS. To capture the TTS starting from the mRNA poly-A tail, a poly-T oligo is used to

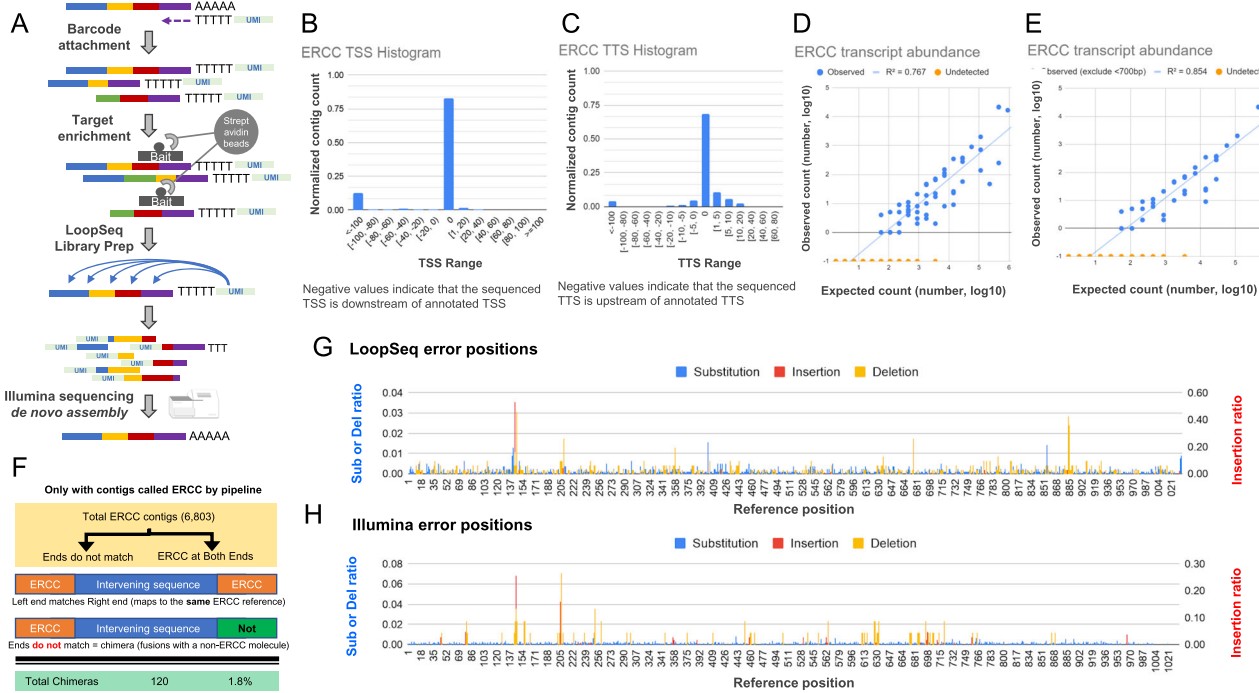

**Fig. 1 Schematics and validation of LoopSeq long-read transcriptome sequencing. A** Overview of the library preparation for isoform sequencing using LoopSeq, including an optional target enrichment step to focus the sequencing depth on genes or isoforms of interest. **B** The transcription start site (TSS) of reconstructed ERCC contigs as compared to the reference annotation. **C** The transcription termination site (TTS) of reconstructed ERCC contigs as compared to the reference annotation. **D** Comparison of ERCC transcript counts between the observed abundance as determined by reconstructed contigs and the expected abundance given the input into the library preparation. **E** Comparison of ERCC transcript counts between the observed and the expected abundance, excluding references <700 bp. **F** Overview of the chimera contig detection. Demonstration of: **G** the positional bias of LoopSeq errors along an ERCC reference and **H** the positional bias of Illumina short-read errors along the same ERCC reference. The ratio of substitution (left axis), deletion (left axis), and insertion errors (right axis) against the position on ERCC-0002 reference are shown. The plotted values are the ratio of each error at a given reference position normalized by the overall error rate of a given error type. The Illumina short-reads used for error analysis are obtained from previously published data[64].

initiate the RT reaction. However, mis-priming can happen when the 3′ end of a poly-T oligo overlaps past the correct poly-A start site due to permissive A/T pairing, offsetting frame alignment, and thus skipping around the last few bases where RT is initiated. Random skipping is consistent with the observed shifts in 15.6% of TTSs that form a normal distribution around the correct, known TTS.

**ERCC transcript quantification.** To evaluate the accuracy of in transcript quantification using LoopSeq, we prepared and sequenced a separate library of 66,308 synthetic RNA control ERCC cDNA molecules, and the observed abundances of ERCC molecules were compared with the expected abundance. As shown in Fig. 1D, the agreement between the observed abundance and the expected expression abundance in the LoopSeq data is comparable to previous reports of the ERCC sample sequenced with previously published SLR and long-read technologies[8]. Similar to what has been reported previously with other long-read technologies such as PacBio sequencing and other SLR methods[8], there are length-related biases in expression quantification. As shown in Fig. 1E, only considering transcripts that are expected to be at least 700 bp in length increases the agreement between the observed and the expected abundance. Taking into account that roughly 68K long-reads were generated for the ERCC sample, it is expected that the ERCC transcripts that dropped out (orange data points on Fig. 1D, E) were the ones at the very low end of the expected ERCC abundance.

**LoopSeq ERCC SLR mis-assemblies/chimeras.** We examined the rate of chimeric sequence formation with large sections of long-reads being incorrect. cDNA synthesis and PCR amplification are known sources of chimeric sequence formation in mRNA sequencing. Reverse transcriptase with template-switching activity has been reported to jump within or between different nucleic acid templates without terminating DNA synthesis activity, resulting in chimeric cDNA formation[13]. Over-amplification during PCR also leads to formation of chimeric molecules[14]. While care is taken to not over-amplify cDNA molecules during PCR, chimeric molecules can sometimes be made during cDNA amplification. LoopSeq employs consensus sequence correction to remove chimeric sequences that are introduced during PCR, but it does not completely eliminate it. If a chimeric molecule is to form, we presume the chimeric junction is likely to occur once in the middle of the molecules, and one of the ends of the molecules would not map to the expected reference. To measure the rate of chimera formation, we examined 6803 reconstructed full-length ERCC contigs and separately mapped the ends to the reference database. As illustrated in Fig. 1F, 120 contigs were found to have ends that do not map to ERCC, which indicates a chimera rate of 1.8%. Most chimeras were formed between molecules (i.e., had a 3′ terminus of molecule A and 5′ terminus of molecule B), not within molecules, which is a hallmark of PCR chimeras. Computational assembly (SPADES) mis-assemblies would be of short-reads that belong to the same molecule, not to two different molecules. This suggests that these mis-assembled contigs are not the result of an assembly error made by SPADES, but the result of

**Table 1 Comparison of error rates across different sequencing platforms.**

| Error type | PacBio-CCS RNA | ONT-2D RNA | Illumina RNA | LoopSeq RNA | LoopSeq DNA |
|---|---|---|---|---|---|
| Match | 9.83E-01 | 8.66E-01 | 9.95E-01 | 9.992E-01 | 9.998E-01 |
| Substitution | 1.3E-02 | 5.50E-02 | 4.15E-03 | 3.475E-04 | 1.476E-04 |
| Insertion | 8.7E-04 | 3.12E-02 | 3.84E-04 | 2.060E-04 | 2.148E-06 |
| Deletion | 3.4E-03 | 4.79E-02 | 4.80E-04 | 2.691E-04 | 2.387E-06 |
| Sum error | 1.72E-02 | 1.34E-01 | 5.018E-03 | 8.226E-04 | 1.521E-04 |

Error rates of PacBio CCS and Oxford Nanopore ONT-2D are obtained from previously published data[63]. Error rate of Illumina short-reads are computed from previously published data[64].

PCR chimeras. Nevertheless, we cannot rule out that some of the erroneous contigs were generated by the assembler, but were misclassified by us as PCR related. A comprehensive evaluation of the de novo assembler used in this work (SPADES) as well as other alternative assemblers has been published elsewhere[15,16].

**LoopSeq ERCC error rates.** To examine LoopSeq's error rate, we compared LoopSeq's contigs to the expected ERCC sequences. A variant table of single-nucleotide edits, either from substitution, insertion, or deletion was constructed by comparing the ERCC full-length contig sequences to the reference sequence of ERCC-00002. As shown in Fig. 1G, plotting the frequency of single-nucleotide variations against the reference shows that while the positions of substitutions are mostly random, the positions of insertions and deletions are concentrated at specific locations. Specifically, 53% of insertion errors were found near position 142, at a homopolymer region of seven As. Substitution errors were also concentrated near homopolymer regions, such as positions 142 (seven As), 203 (five As), 672 (four As), and 882 (four Ts). To examine whether substitution errors originated from LoopSeq sequencing, we performed the same variant sequence analysis on a previously published ERCC Illumina short-read dataset. As shown in Fig. 1H, the published ERCC Illumina dataset and the LoopSeq dataset share some of the most abundant mismatches to the published reference ERCC sequences, including positions 142 and 203, the most abundant mismatches in the entire dataset. This suggests that these highly abundant mismatches may be errors introduced during synthetic RNA synthesis, since they are shared by both methods. Besides these several abundant mutations that are shared between the two datasets, there was no statistically significant correlation between the mismatched position in the Illumina and LoopSeq datasets for any of the error categories (Supplementary Data 1). This result is in line with our expectation that some mutations (a small minority of the errors) should be shared by both methods due to errors in the starting RNA material while others should be either random or method-specific due to the differences in sample preparation methodologies.

Improvements to the uniformity of short-read coverage (Supplementary Fig. S1) along the length of barcoded long transcripts have resulted in a lowering of the error rates in LoopSeq SLRs compared to previously published SLR methods. Uniform short-read coverage along barcoded molecules helps reduce the error rate by avoiding low-coverage regions, which have less rigorous consensus-based error correction. While this study quantifies the gains in reducing the error rates compared to PacBio and ONT, making a similar comparison to other SLR methods is challenging because previously published SLR technologies[10,11,17,18] did not evaluate error rates with ground truth RNA samples. Tilgner et al., 2015[8] (which used TruSeq SLRs) is the only previously published SLR ERCC data, but it focused on observed versus expected expression, not error rate analysis, thereby making it difficult to systematically compare their error rates to LoopSeq error rates. Specifically, Tilgner et al.

did not report the mismatch rate (the largest source of sequence error), the standard manner for reporting error rates as reported for PacBio, ONT, and LoopSeq here. Furthermore, Tilgner et al. did not address their method's inability to distinguish between highly homologous isoforms within the same sample, which is expected to increase the error rates in real samples that contain isoforms.

We performed a comparative analysis of error rate between LoopSeq, PacBio CCS reads, Oxford Nanopore 2D reads, and Illumina short-read sequencing. Table 1 summarizes the error rates across the different sequencing methods. When sequencing synthetic RNA, LoopSeq exhibits at least two orders of magnitude lower error rate compared to PacBio CCS reads or Oxford Nanopore 2D reads, and an order of magnitude lower error rates compared to Illumina short-read sequencing. As suggested by the non-random nature of the indel errors observed in the LoopSeq RNA sequencing data, the errors are not characteristic to the technology but rather to how the RNA is converted to cDNA using template switching. When sequencing DNA templates directly without reverse transcription[12], LoopSeq's indel errors are reduced to a negligible rate, roughly 1 in 220,000 bases assembled.

Finally, we devised a quality score model (Supplementary script under 'Q score model') that assigns a quality score for each bp in each SLR and writes the SLR data in standard FASTQ format. The quality score model integrates (1) the number of bp's from independent short-reads covering every position in the SLR, (2) their Q scores from the shprot-read FASTQ files, and (3) the degree of consensus between them per position in the SLR.

**Quantification of gene and isoform expressions in human colon cancer.** To examine the utility of LoopSeq in quantifying the expressions of genes and isoforms in human cancers, we prepared and sequenced three pairs of primary colon cancers and their matched metastases in the lymph nodes. To enrich cancer-specific gene fusions and isoforms, a panel of 2149 probe-capture oligos representing 2193 genes were designed (Supplementary Data 2). These oligos represent the split regions of the most frequent cancer-related gene fusions found in the TCGA databases. As LoopSeq can identify transcripts at isoform resolution, we were able to quantify the expression at both gene-level and isoform-level. Using the fusion-junction probe-capture oligos, we obtained transcripts of 12,127 cancer-related genes from the cancer samples (Supplementary Data 3). As shown in Fig. 2A, we found 2682 genes that were differentially expressed between the metastasis and primary cancer samples or between the tumor samples and their corresponding benign colon tissue adjacent to cancer (Supplementary Data 4). When hierarchical clustering analyses were performed, differentially expressed genes (DEGs) showed proper segregation between primary cancer/metastases and normal colon samples (Fig. 2A and Supplementary Data 4). However, the separation between primary cancer and metastasis samples is inadequate, and we were unable to differentiate between primary cancer and metastasis samples with gene

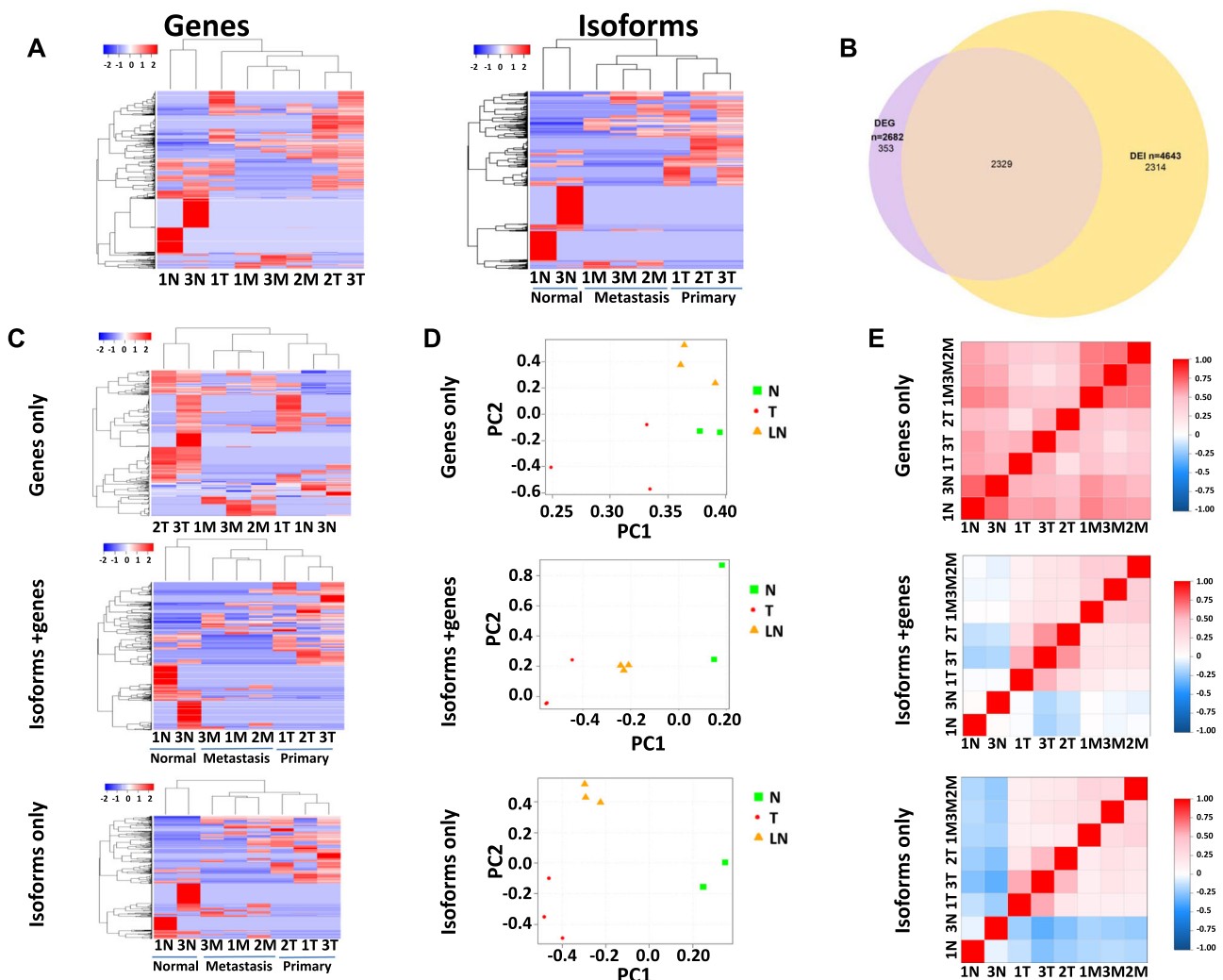

**Fig. 2 Tissue segregation by cancer progression stage using isoform-level versus gene-level expression data. A** Hierarchical clustering of benign colon samples adjacent to cancer (1 N and 3 N), primary colon cancer samples (1–3 T) and metastatic colon cancer samples (1–3 M) based on differentially expressed genes (left) or isoforms (right). The color reflects the indicated-row Z score. **B** Venn diagram of overlapping differentially expressed genes and isoforms in colon cancers, metastases, and benign colon tissues adjacent to cancer. **C** Hierarchical clustering of colon samples based on differential expressed genes but not isoforms (top), or differential expressed genes accompanied with concomitant isoform differential expression (middle), or different isoform expressions without alteration of gene expression (bottom). The color reflects the indicated-row Z score. **D** Principal component analyses of benign colon tissues adjacent to cancer, primary colon cancers, and metastatic colon cancers based on differential gene expression without isoform expression alteration (top), or differential gene expression with concomitant isoform alteration (middle), or differential isoform expression without the alteration of gene expression (bottom). **E** Pearson's correlation of benign colon tissues adjacent to cancer, primary colon cancers, and metastatic colon cancers based on differential gene expression without isoform expression alteration (top), or differential gene expression with concomitant isoform alteration (middle), or differential isoform expression without the alteration of gene expression (bottom). The color reflects Pearson's correlation coefficient for the pairing samples.

expression patterns alone. When leveraging the isoform mapping from reconstructed long-reads, we found 5941 differentially expressed isoforms (DEIs) across 4643 unique genes among these samples (Supplementary Data 5). Unlike DEGs, DEIs showed excellent segregation of all three groups of tissues (Fig. 2A and Supplementary Data 5), demonstrating the power of performing differential expression analysis using isoform expression data over gene expression data.

To further elucidate the significance of gene versus isoform expression patterns, next we examined DEGs with no isoform expression change and DEIs with no gene expression changes. The Venn diagram in Fig. 2B illustrates the overlapping between DEIs and DEGs at gene-level. While only 13.3% (353 of 2682) DEGs have no change in isoform distribution (Fig. 2B), nearly half (49.9% or 2316/4643) of DEIs belong to genes with no gene

expression changes (Supplementary Data 6). To investigate the impact of DEG with no change in isoform distribution on tissue differentiation, a hierarchical clustering analysis was performed on the primary colon cancer, the lymph node metastasis, and their matched benign colon tissues adjacent to cancer using 353 DEGs that have no isoform alterations. The results indicated that these DEGs are not able to segregate the normal samples from the cancer samples (Fig. 2C and Supplementary Data 4 and 6). Both principal component and Pearson's correlation analyses confirmed the inadequacy of tissue segregation based on these genes (Fig. 2D, E). In contrast, clustering analysis using DEGs with isoform redistribution produced a far better segregation between the benign colon and cancer samples (Fig. 2C–E, mid panel and Supplementary Data 4–6). Interestingly, isoform redistribution without a change in gene expression produced the best tissue-

differentiation results: all benign colon, primary cancer, and metastases samples were segregated into different groups (Fig. 2C, lower panel and Supplementary Data 5 and 6). These results indicate that DEI analysis produced robust sample segregation not obtainable by DEG analysis. These DEIs, which might have previously been inaccessible and were hidden within comparable gene expression levels, represent an additional dimension in differential expression analysis.

Genes involved in cancer metastasis such as *CD44*[19] are among the DEIs that are not accompanied with a change in gene expression (Supplementary Fig. S2). There were 26 different isoforms of *CD44* detected in the colon cancer tissues, with the protein length ranging from 139 to 743 aa. Isoform analysis indicates that 2 isoforms (XM_005253231.3 and XM_011520484.2) emerged in the colon cancer and cancer metastasis samples but were absent in the benign colon tissues. Another gene of interest is *ATP1A1*, a Na$^+$/K$^+$ ATPase that is a subunit of Na$^+$/K$^+$ pump essential for maintaining ionic homeostasis for a cell[20]. *ATP1A1* produced 2 additional isoforms (NM_000701.7 and NM_001160233.1) in the primary colon cancer samples and their corresponding metastasis, but not in the benign colon tissues (Supplementary Fig. S3). The distribution of *ATP1A1* isoforms was validated by Taqman qRT-PCR (Supplementary Fig. S4).

**Expression pattern analyses of isoform-specific single-nucleotide variants.** Mutation is the hallmark of human malignancies. However, little is known about isoform-specific mutations due to the difficulty in identifying mutations and isoforms simultaneously. Taking the advantage of the read-length and the accuracy of LoopSeq SLRs, we examined the single-nucleotide variants (SNVs) in assembled contigs in the context of isoforms. A total of 4042 SNVs was identified in 6 cancer samples using LoopSeq that were cross-validated by standard short-read whole-exome sequencing (WES; Supplementary Data 7). These SNVs were distributed among 1340 genes and 8712 isoforms. Interestingly, many SNVs were found in specific isoforms for a given gene. Of the 1509 SNVs found with at least 2 isoforms and 5 assembled contigs, 1297 SNVs were not distributed evenly among the isoforms of these genes but were predominantly found in specific isoforms (Supplementary Data 8). While the majority of the SNV isoform distribution is comparable to the wild-type isoform distribution, the isoform expression patterns of 113 SNVs were significantly different from their wild-type counterparts (Supplementary Data 9), suggesting the alterations of splicing patterns for the variants. To validate the SNV isoform expression patterns observed in the long-read data, isoforms of two genes were selected for targeted and isoform-specific short-read sequencing, made possible by the close proximity of the mutation and the alternative-splicing junction. For *FAM104A*, a protein involved in centriole biogenesis[21], the SNV isoform expression pattern is comparable to the wild-type counterpart of NM_001098832, NM_032837, and NM_001289410, and the short-read data were largely consistent with the LoopSeq data (Supplementary Data 10A). For *PABPC1*, a poly-A binding protein[22], the SNV isoform distribution of NM_002568 and XM_005250861 was different from the wild-type distribution, and again similar observation was made with the short-read data (Supplementary Data 10B).

To identify SNVs that change their expression in a given isoform during the progression of colon cancers, we screened for isoforms that uniformly have high SNV rate (≥ 0.5) or low SNV rate (≤ 0.5) across all metastatic colon cancer samples versus matched primary cancer samples. SNV rate was computed by normalizing the SNV counts with the total transcript counts of an isoform. Twenty-three SNV-containing isoforms were identified

that match the search criteria (Supplementary Data 11). The hierarchical clustering analysis based on the SNV rates of 23 SNV-containing isoforms confirmed that these isoforms produced a complete separation between cancer and metastatic samples (Fig. 3A and Supplementary Data 11). Similar results were obtained by the principal component and Pearson's correlation analyses (Fig. 3B, C). In contrast, hierarchical clustering analysis based on the SNV rates of all 8712 isoforms failed to yield appreciable separation of metastatic and primary cancer samples (Supplementary Fig. S5 and Supplementary Data 7). The ingenuity pathway analysis indicates that many of the 23 SNV-containing isoforms belong to genes involved in DNA-repairing signaling and antigen-presentation signaling (Fig. 3D).

To study the potential pathological SNVs in the development of colon cancer, we cross-referenced the 4042 validated SNVs against the database of Catalogue of Somatic Mutations in Cancers (COSMIC). In all, 401 SNVs were identical to the mutations in the colon cancer database of COSMIC, suggesting many SNVs we discovered in our dataset may be involved in the pathological progress of colon cancer (Supplementary Data 12). Of these potential pathological SNVs, 190 were present predominantly in some specific isoforms of the residing genes. Twelve SNVs displayed different isoform expression patterns in comparison with their wild-type patterns. Two SNV examples we investigated are *BRAF* and *KRAS*. The V600E variant[23] of *BRAF* was detected in sample 3 T (primary tumor sample) and 3 M (metastasis tumor sample). The predominant isoform for both wild-type and the V600E variant was NM_04333 in both samples ($p = 0.006$ for 3 M WT, $p = 1.6E-7$ for 3 M V600E, $p = 0.034$ for 3 T WT, and $p = 1.7E-21$ for 3 T V600E). However, additional isoforms for *BRAF* (NM_001354609, XM_017012558, and XR_001744857) emerged in both primary cancer and the corresponding metastasis, and all contained the V600E variant (Fig. 4). In contrast, wild-type isoform distribution of *KRAS* proto-oncogene[24] is distinct between samples 2 T (primary tumor sample) and 2 M (metastasis tumor sample). Similarly, the G12V variant also had different isoform distributions. This suggests that *KRAS* may have undergone a change in isoform distribution when the colon cancer evolved from its primary site (2 T, NM_033360) to the lymph node site (2 M, NM_004985 and XM_011520653) (Fig. 4).

**Discovery of fusion gene isoforms.** Discovery of fusion isoforms remains difficult due to the requirement that the fusion junction needs to be sequenced alongside exons that can differentiate one isoform from another. To search for the known fusion transcripts, we first performed text-searching to identify the fusion transcripts reported by TCGA (Supplementary Data 2) in our samples. We then used SQANTI[25], a bioinformatics pipeline for classifying long-reads by splice-junctions, to detect fusion transcripts in the SLR data. We applied two additional filtering criteria on the fusion isoform candidates: (i) the fusion gene partners are of trans direction or cis direction separated by >40 kb with at least one gene in between, and (ii) the fusion-junction point is derived from the exon junctures between the fusion partners (Supplementary Fig. S6). Among the 6 samples of colon cancers, 4 previously unknown fusion isoforms were found to meet these criteria. These fusion junctions were additionally confirmed by Minimap2[26,27] and STARlong[28] aligner (Supplementary Data 13), and validated using Taqman qRT-PCR and Sanger sequencing (Fig. 5 and Supplementary Figs. S7–S11).

STAMBPL1-FAS fusion isoform was identified in sample 1 M (metastasis cancer) and 2 M (the metastasis cancer): XM_011539985 from *STAMBPL1* and XM_011539766.2 from

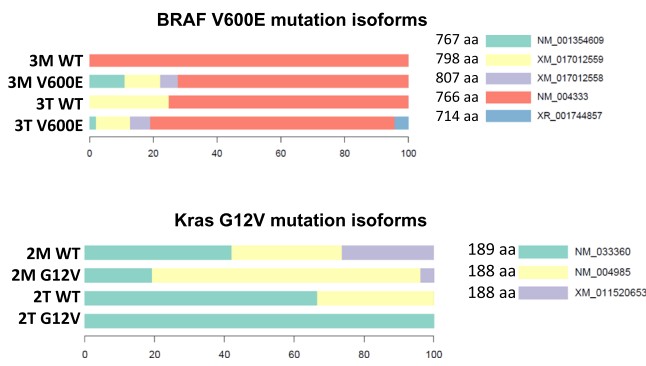

**Fig. 3 Isoform switches of single-nucleotide variant between primary colon cancers and metastatic colon cancers. A** Hierarchical clustering between primary colon cancers and metastatic colon cancers based on the quantities of non-synonymous SNVs of 23 isoforms in each sample. The color reflects SNV rate by fraction. **B** Principal component analyses of primary colon cancers and metastatic colon cancers based on the quantities of non-synonymous SNVs of (**A**). **C** Pearson's correlation of primary colon cancers and metastatic colon cancers based on the quantities of non-synonymous SNVs of (**A**). The color reflects Pearson's correlation coefficient for the pairing samples. **D** Pathway analysis of 23 single-nucleotide-variant isoforms showed enrichment in genes involved in HLA/CD74 antigen presentation pathways.

**Fig. 4 Mutant isoform switching of *BRAF* V600E and K-ras G12V in colon cancers.** Top panel: Isoform distribution of V600E of *BRAF* in colon cancer. Bottom panel: Isoform distribution of G12V of *KRAS* in colon cancer.

*FAS* (Fig. 5A). However, subsequent analysis using Taqman qPCR showed that the *STAMBPL1-FAS* fusion isoform can be found in all 6 cancer samples (Supplementary Fig. S7), implying a wider distribution of this gene fusion in colon cancers. *STAMBPL1* is a

deubiquitinase involving NF-kB activation and the inhibition of apoptosis[29]. *FAS*, on the other hand, is a cell surface death receptor[30]. The *FAS* isoform (XM_011539766.2) in the fusion protein contains the transmembrane domain and death domain in the cytoplasmic portion of the protein. The fusion isoform is a chimera of the largely intact metalloprotease domain from *STAMBPL1* fused with the extracellular domain of *FAS*. Since *FAS* is a transmembrane protein while *STAMBPL1* is an endosomal one, it is unclear where the ultimate subcellular localization of the chimera protein is or whether the activity of *FAS* is neutralized by *STAMBPL1*. In contrast, both the *PTPRK-ECHDC1* gene fusion isoform and the *ZNF124-SMYD3* gene fusion isoform introduced a frame-shift by the fusion event (Fig. 5B, C and Supplementary Fig. S8). As a result, only the truncated *PTPRK* and *ZNF124* proteins are produced. The tail gene expressions are eliminated (Supplementary Fig. S9). *GNAS* is a component of the guanine nucleotide binding protein[31], and is frequently mutated in colon and pancreatic cancers[32]. In our analysis, *GNAS* forms a fusion gene with *VAPB*[33] (Fig. 5D and Supplementary Fig. S10). The fusion generated a chimeric protein resulting in a loss of the regulatory domain in the N-terminus of *GNAS* while leaving the G-binding domain intact. The chimera protein may have a previously unknown function in the cancer cells.

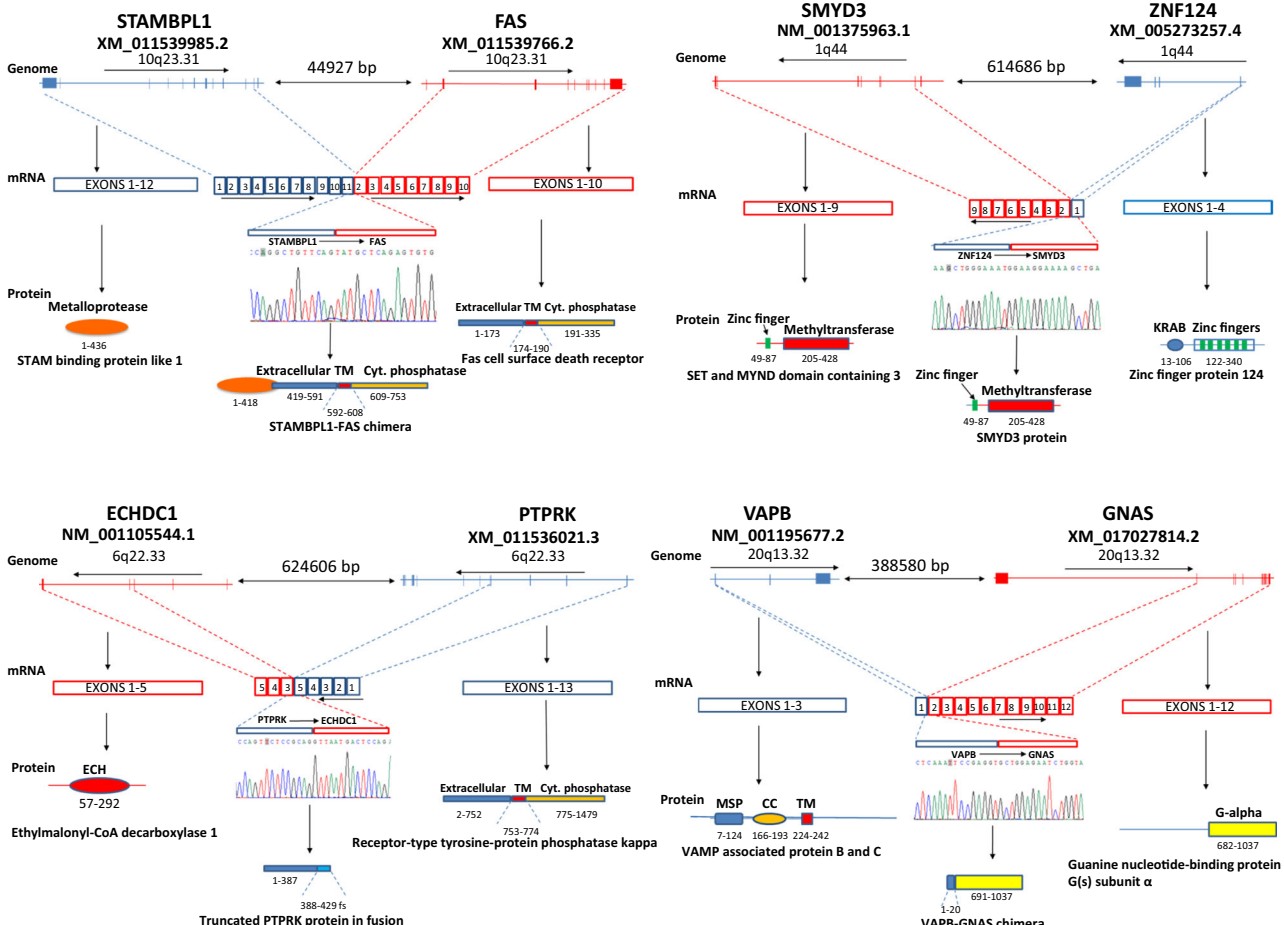

**Fig. 5 Validation of previously unknown fusion gene isoforms identified in colon cancers through LoopSeq sequencing. A** STAMBPL1-FAS fusion. Top: diagram of mini genomes of STAMBPL1 and FAS. Direction of transcription and distance between the two genes are indicated. Middle: mRNA represented by exons from each gene. Bottom: diagram of functional protein domains of STAMBPL1 and FAS. **B** ZNF124-SMYD3 fusion. Top: diagram of minigenomes of ZNF124 and SMYD3. Direction of transcription and distance between the two genes are indicated. Middle: mRNA represented by exons from each gene. Bottom: diagram of functional protein domains of ZNF124 and SMYD3. **C** PTPRK-ECHCD1 fusion. Top: diagram of minigenomes of PTPRK and ECHCD1. Direction of transcription and distance between the two genes are indicated. Middle: mRNA represented by exons from each gene. Bottom: diagram of functional protein domains of PTPRK and ECHCD1. **D** VAPB-GNAS fusion. Top: diagram of minigenomes of VAPB and GNAS. Direction of transcription and distance between the two genes are indicated. Middle: mRNA represented by exons from each gene. Bottom: diagram of functional protein domains of VAPB and GNAS.

## Discussion

The survival of eukaryotic cells requires rapid adaptability to the constant changes of their environment. By utilizing different combinations of exons of a gene, eukaryotic cells generate an array of different isoforms and proteins from a single gene, which allow them to cope with the challenge of the environment[34]. Even though alternative splicing of mRNA has long been known and is robust in most genes, little is known about the isoform distribution in cells. Isoform quantification has been hampered by the complexity of alternative splicing and the lack of adequate tools to identify the different isoforms. In previous studies, long-read sequencing technologies from Pacific Biosciences (PacBio) or Oxford Nanopore (ONT) were used to sequence full-length transcripts, either on cDNA synthesized from RNA or directly on RNA molecules[35]. However, the single long-molecule readout offered by PacBio SMRTseq must be accompanied by short-read RNAseq for error correction, effectively requiring independent rounds of sequencing per sample. Without short-reads for correction, downstream informatics processing is needed, with varying levels of error reduction[36,37]. ONT has native RNA readability, skipping the need for cDNA library construction, but

is hampered by low throughput, high error rate, and incomplete read lengths[38,39]. The relatively high error rates of both long-read sequencing methods hamper their usefulness in accurate isoform mapping and quantification. In comparison, LoopSeq combines the low error rate of Illumina short-reads with long-molecule data that facilitates transcriptome profiling and isoform discovery. Short-read densities that cover each contig/UMI allow for error correction by base-pair consensus. Additionally, UMI tagging enables accurate assessment of relative transcript abundance. Our results demonstrate that isoform characterization with LoopSeq enables obtaining detailed granularity in isoform expression regulation, isoform-specific mutation expression, and fusion gene isoform expression that were previously inaccessible.

Comprehensive quantification of isoform-specific mutations were seldom performed, even though mutations at the genome levels have been extensively studied in human cancers. Our analysis showed that most mutations are not evenly distributed among the isoforms, but rather are dominantly present in some isoforms. A significant number of mutations undergo a change in isoform distribution as the cancer evolves. These changes in the expression of dominant mutation isoform may bear important

clinical significance, as it may alter signaling pathways and adapt cancer cells to their new environment. Lastly, drug targeting design relies on accurate assessment of the physical interaction between the drug and the structure of the protein target. Subtle variations in the amino acid sequences among different protein isoforms can have a profound impact on the interaction between the drug and its targets. Isoform switching can also impact the signaling mechanism of cancer-driver proteins, leading to resistance to cancer treatment[40,41]. The ability to accurately characterize and quantify isoforms of target proteins will undoubtedly provide additional insight in cancer-drug design and shed light into the mechanism of cancer-drug resistance.

## Methods

**Samples collection for internal quality control**. ERCC synthetic RNA and Hela total RNA were obtained from ThermoFisher Scientific, Inc., and used as internal quality control to demonstrate the error rate, chimera rate, and transcript quantification using LoopSeq.

**Colon cancer sample collection and RNA extraction**. Frozen tissue samples were collected from 3 colon cancer patients, including benign colon tissue adjacent to cancer samples, primary colon cancer samples, and lymph node metastasis samples. The procedure of obtaining the tissues and informed consent exemption were approved by the institutional review board of University of Pittsburgh. The tissues were fresh-frozen in liquid nitrogen, and stored in −80 °C. Total RNA was extracted using TRIzol (Invitrogen, CA) and RNeasy column methods[42–51]. Briefly, tissues were homogenized in a homogenizer. The homogenized tissues were lysed with TRIzol/chloroform (5/1 ratio). After a brief centrifugation, the aqueous phase of the lysate was incubated with isopropyl alcohol (1 mL aqueous phase per 0.5 mL isopropyl alcohol) for 20 min at room temperature. The RNA was then precipitated by centrifugation at 12,000g for 30 min. The RNA was then washed with 70% alcohol and dried in a SpeedVac$^{TM}$.

**Target sequence selection for loop sequencing**. Fusion transcript candidates were selected from University of Pittsburgh Medical Center (UPMC) cohort and TCGA database. For the UPMC cohort, 14 fusion transcripts were detected by our previous study[47,49]. For the TCGA panel, a list of 17,754 fusion transcripts were downloaded from Tumor Fusion Gene Data Portal (https://www.tumorfusions.org/). Based on the Cancer Gene Census (https://cancer.sanger.ac.uk/census), 315 oncogenes and 315 tumor suppressor genes (TSGs) were defined. Eventually, 2135 fusions were selected by satisfying one of the following criteria: (1) fusions can be detected in more than one sample; (2) two genes involved in one fusion transcript are either oncogenes or TSGs; (3) only one fusion gene is an oncogene or TSG, and the fusion event is either in-frame or out-of-frame. To sum up the two cohorts, in total 2149 fusion transcripts were selected (Supplementary Data 2), which involved 2193 unique genes. The 100 bp sequences surrounding the fusion-junction point (50 bp upstream to the junction and 50 bp downstream to the junction) were extracted and used for enrichment. The oligonucleotides were provided by Twist Bioscience, Inc., CA.

**LoopSeq sequencing library preparation**. LoopSeq sequencing libraries were prepared from ERCC synthetic RNA, a blend of ERCC synthetic RNA and Hela total RNA, or total RNA extracted from tissue samples using LoopSeq SLR Transcriptome Kit according to manufacturer's protocol and previously described SLR chemistry[11], except when specified. Specifically, 200 ng of total RNA extracted from cancer tissues was reverse-transcribed and barcoded using an UMI-containing barcoding primer that primes on the poly-A tail LoopSeq Transcriptome Kit. The number of barcoded cDNA molecules is determined by the efficiency of priming on the poly-A tail and the efficiency of template switching on the 3′ terminus of the cDNA. As such, the number of barcoded transcripts that can be generated is comparable to previously published methods that employ a poly-A-priming and template-switching strategy for generating cDNA. Subsequently, the barcoded cDNA was enriched using double-stranded oligonucleotide probes (Twist Biosciences, CA) targeting 2149 fusion-junction sequences (selected in previous step) prior to the amplification of full-length cDNA molecules (Barcode Amplification in LoopSeq library preparation). Specifically, the barcoding reaction was purified using 0.6X SPRIselect ratio and then eluted into 20 μL of Hybridization Mix per manufacturer's pre-capture concentration protocol. The hybridization reactions contain 20 μL of Hybridization Mix, 5 μL of Blocker Solution, 9 μL of LG Adaptor Blocker (Loop Genomics), 4 μL of Twist Custom Enrichment Panel, 2 μL of Buffer EB (Qiagen), and 30 μL of Hybridization Enhancer. All probe hybridization and washes were conducted per manufacturer's protocol. All target enrichment reagents are from Twist Biosciences except when specified. The captured barcoded cDNA was amplified following the LoopSeq Transcriptome protocol starting from Barcode Amplification. Briefly, the captured barcoded cDNA (2 μL) was mixed 15 μL Amplification Mix R and 3 μL of Amplification Additive,

and amplified in the following condition: 95 °C for 5 min, followed by 14 cycles of 98 °C for 20 s, 60 °C 20 s and 72 °C 10 min. After the clean-up with SPRIselect (Beckman Coulter), the amplified barcoded cDNA (15 μL) was mixed 2 μL Distribution Enzyme and 5 μL Distribution Mix, and underwent 1 cycle of 20 °C for 15 min and 75 °C 5 min for barcode distribution. The barcode activation was achieved by adding 75.5 μL Activation Mix and 2.5 μL Activation Enzyme to the barcoded cDNA from the previous step under the following condition: 20 °C for 16 h and 65 °C 10 min. Barcode neutralization was obtained by adding 6 μL Neutralization Enzyme in 37 °C for 15 min. After SPRIselect clean-up, the barcode-distributed cDNA from the previous step (20 μL) was fragmented by adding 20 μL Fragmentation Mix and 10 μL Fragmentation Enzyme for 1 cycle of 32 °C for 5 min and 65 °C 30 min. The fragmented cDNA (50 μL) was then ligated by adding 40 μL Ligation Mix and 10 μL Ligation Enzyme at 20 °C for 15 min. Following the clean-up, the DNA (20 μL) was amplified and indexed by adding 25 μL Index Master Mix and 5 μL Index Primer, and underwent the following condition: 1 cycle of 98 °C for 45 s, followed by 11 cycles of 98 °C for 15 s, 60 °C 30 s, and 72 °C 30 s. The indexed and fragmented cDNA will be cleaned up with SPRIselect and assessed for the quantity and quality through Qubit and Bioanalyzer 2000 evaluation before Illumina sequencing.

**Probe hybridization analysis**. Selected bases' on-target rate is computed by aligning the contigs to GRCh37 reference genome using minimap2[26] and comparing the alignment against the probe BED file using Picard. As shown in Supplementary Data 14, the on-target reads of the transcriptome sequencing ranged from 83% to 92% in long-read level, and 93% to 96% in base level per sample, indicating a high level of enrichment by the probes. In addition, two known fusion genes (AGRN-NOC4L and CTNND1-TMX2) in the cancer sample (3 T) were enriched and detected in the dataset, suggesting that the probe design for the fusion transcript is adequate.

**Isoform expression analysis**. Transcriptome samples of 8 samples across 3 colon cancer patients were measured by LoopSeq technology, including 2 benign colon tissue samples (1 N and 3 N), 3 tumor samples (1 T, 2 T, and 3 T), and 3 metastasis samples (1 M, 2 M, and 3 M). LoopSeq long-molecules were analyzed by tool SQANTI for transcriptome isoform identification and quantification[25]. Isoforms were quantified by read count across all the 8 samples. Differential expression analysis was performed by R tool 'DEseq2'[52] to compare normal samples versus tumor samples, and tumor samples versus metastasis samples, respectively. Top DEIs were selected by false discovery rate FDR = 5% and absolute fold change greater than 2-fold. Then hierarchical clustering[53] was applied on these 8 samples based on the DEIs pooling from the two comparisons. Besides isoform-level analysis, similar analysis was performed at gene level to detect DEGs and cluster samples based on these DEGs. To compare isoform-level and gene-level analysis, these DEIs/DEGs were categorized into three groups: DEG only, DEG/DEI intersect, and DEI only. Hierarchical clustering, principal component analysis (PCA), and Pearson correlation analysis were performed based on genes/isoforms within each category.

**SNV isoform analysis**. Loop-seq long-reads were aligned to the human reference genome hg38 by Minimap2 aligner[26]. For each sample, mutations/SNPs were called and quantified by SAMtools mpileup function[54,55]. These mutations/SNVs were then annotated by ANNOVAR tool, dbSNP, and Cosmic database to identify known SNPs and somatic mutations in human cancer[56–58]. Those detected SNVs were further filtered by the following criteria: (1) validated by WES (see next section) and (2) either stop-gain or non-synonymous mutations/SNVs. For a given SNV position of interest, reads with reference base (wild-type) and alteration base were able to be identified and annotated at isoform level by SQANTI[11].

Several statistical tests were applied to the SNV isoform data. (1) When defining the unevenly distributed SNV isoforms per gene, only SNVs involved with more than one isoforms and covered by at least 5 contigs were analyzed by Chi-squared test. p-values were adjusted by Benjamini-Hochberg (BH) method and FDR = 5% were applied to define significant unevenly distributed SNV isoforms. (2) When detecting differentially expressed SNV isoforms between reference (wild-type) and altered alleles, Fisher's exact tests were applied to test the long-molecule read count of reference/altered alleles across multiple SNV isoforms. BH adjustment and FDR = 5% were applied. (3) When comparing tumor samples and metastasis samples, SNV rate was defined as SNV count divided by total count (SNV count + wild-type count). Based on this, SNV isoforms with SNV rates low in tumor samples and high in metastasis samples were detected, or vice versa. Hierarchical clustering, PCA, and Pearson correlation analysis were applied on these selected switching isoforms. We further applied these top switching isoforms for IPA pathway analysis (QIAGEN Inc., https://www.qiagenbioinformatics.com/products/ingenuitypathway-analysis). Significant pathways were visualized by network plot drawn by Cytoscape[59].

**Single-nucleotide variant calling from WES as validation**. WES was performed on the same three individuals for mutation validation at DNA level. Illumina TruSeq Exome kit was used to prepare the exome DNA libraries of 2 T, 2 M, 1 T, 1 M, 3 T, and 3 M. The genome DNA was sheared to 150 bp using Covaris

sonicator. The fragmented DNA was end-repaired, polyadenylated, and ligated with Illumina adapters. The adapter-ligated DNA was amplified by PCR for 8 cycles in the following condition: 98 °C for 20 s, 60 °C for 15 s, and 72 °C for 30 s. The amplified libraries were then pooled and bound with Coding Exome oligos. The hybrids were then captured by Streptavidin Magnetic Beads provided by Illumina, Inc. The beads were then washed. The captured libraries were eluted. The capture procedure was repeated once. The eluted libraries were amplified for 8 cycles under the same condition as above. The libraries were cleaned up and assessed for quantity and quality based on Agilent's Bioanalyzer 2000 and Qubit. The libraries were sequenced on a Illumina NextSeq Dx550 sequencer.

For raw sequencing reads pre-processing, quality control was applied by tool FastQC (https://qubeshub.org/resources/fastqc). Low-quality reads and adapter sequences were trimmed by tool Trimmomatic[60]. Surviving reads were then mapped to human reference genome hg38 by Burrows-Wheeler Aligner[61]. Aligned reads were sorted and marked duplicates by tool Picard (http://broadinstitute.github.io/picard). Mutation/SNP calling on individual samples were performed by SAMtools mpileup function[54,55].

**Amplicon sequencing validation of mutation isoform expression**. Transcriptome sequencing on the same samples as loop-seq were performed for mutation isoform validation. Amplicon sequencing was specifically targeted on two candidate genes: *FAM104A* and *PABPC1* using primer sets ACAACCCCCTCTG TTCCCTCT/ATGGTCTGGCTCAAGCTGCCT for *FAM104A* and AGCAAATGT TGGGTGAACGGC/TTCTTCGGTGAAGCACAAGTTTC.

For bioinformatics analysis, raw sequencing reads first went through the pipeline of quality control by FastQC and then low-quality reads and adapter sequences were filtered out by tool Trimmomatic[60]. Surviving reads were then aligned to human reference genome hg38 by HISAT2 aligner[62]. Mutation calling was performed by SAMtools mpileup function[54,55] and isoform identifications were supported by the reads exactly splitting across more than one exon and their counterpart paired-ends spanning.

**Taqman qRT-PCR to validate isoform expressions**. Two microgram of RNA was used to synthesize first-strand cDNA with random hexamer primers and Super-script II™ (Invitrogen, CA). For NM_000701.8//XR_002956654.1 of ATP1A1 detection, 1 μL of each cDNA sample was used for TaqMan PCR with 50 heating cycles at 94 °C for 30 s, 61 °C for 30 s, and 72 °C for 30 s using the primers and probes listed in Supplementary Data 15 (Primers and probes design). At least one negative control and a synthetic positive control were included in each reaction batch. The PCR products were gel purified, and Sanger sequencing was performed on the positive samples. The procedure of fusion gene validation also followed the similar process except using the primers and probes listed in Supplementary Data 15 (Primers and probes design).

**Fusion transcript detection**. Fusion transcripts were detected from the LoopSeq long-molecules by two methods. For the first method, 2149 targeted fusion junctions (Supplementary Data 2) were specifically checked. Text search based on the 15 bp ahead of and 15 bp right after the junction point were applied to the long-read molecules. An R language script for the high-throughput searching on 20K+ fusions was attached in the Supplementary script files. Once fusion candidates were detected, we extended 15 bp + 15 bp to the full long-read and applied NCBI BLAST tool to perform the alignment to confirm whether the candidate long-read was a true positive. For the second method, SQANTI tool[25] was applied to detect novel fusion isoforms. Supporting reads were additionally confirmed by aligner minimap2[26,27] and STARlong[28].

**Statistics and reproducibility**. All the bioinformatics processing was performed by bash command on the Unix/Linux system. All the downstream statistical analysis was performed by R programming with available packages. $p$-value ≤ 0.05 cutoff was used to define significance. For multiple hypothesis testing, adjusted $p$-value ≤ 0.05 (FDR = 5%) was used to define significance.

For the experimental design, 8 colon cancer samples across 3 patients were analyzed, with 2 normal samples, 3 primary cancer samples and 3 metastasis lymph node samples. Samples across 3 patients will be regarded as replicates in this study to increase the robustness and reproducibility of the results.

**Reporting summary**. Further information on research design is available in the Nature Research Reporting Summary linked to this article.

## Data availability
The detail data for Fig. 1 are located in Supplementary Data 1. The detailed data for Fig. 2 are located in Supplementary Data 4–6. The detail data for Figs. 3 and 4 are located in Supplementary Data 8, 9, 11, and 12. The detail data for Fig. 5 are located in Supplementary Data 13. Data for LoopSeq quality control, LoopSeq colon cancer samples, and RNA amplicon sequencing were submitted to the GEO database, which can be accessed by GSE155921. Raw whole-exome sequencing data for colon cancer samples were submitted to the SRA database with Bioproject accession number PRJNA648918. All the other relevant data are available from the corresponding authors upon request.

## Code availability
All the scripts were summarized in Supplementary Note 1 with detailed parameter setting for the reproducibility of the results. Scripts were uploaded to Zenodo with https://doi.org/10.5281/zenodo.4563137.

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

## Acknowledgements

This research was supported in part by the University of Pittsburgh Center for Research Computing through the resources provided. The work was supported in part by grants from National Cancer Institute (1R56CA229262-01 to J.H.L.), Department of Defense (W81XWH-16-1-0541 to J.H.L.), and University of Pittsburgh Medical Center (Endowed Chair of Molecular Carcinogenesis).

## Author contributions

J.H.L. and T.B.Y. conceived the idea and oversaw the study. B.R., Y.P.Y., and I.W. conducted the experiments. S.L., I.W., and M.B. conducted the bioinformatics analysis.

## Competing interests

The authors declare the following competing interests: I.W., M.B., and T.B.Y. are employees of Loop Genomics, Inc. S.L., Y.P.Y., B.R., and J.H.L. declare no competing interests.
