## [Peer Review File · Communications Biology]

Reviewers' comments:

Reviewer #1 (Remarks to the Author):

This manuscript describes a method that uses a barcoding-based synthetic long read sequencing to identify expressed isoforms from a transcriptome. The authors present it to be more accurate than alternative methods of identifying isoforms, and they apply it to samples from colon to identify isoform switching expression and specific SNVs in cancer progression. This is a noteworthy result, if it wasn't identified before.

My main concerns with this manuscript are related to the description of the LoopSeq method and its accuracy, and are detailed below:

1. It is not clear how LoopSeq is different from previous methods. The authors cite a couple of methods that were previously developed, and they specify "building upon" these previous methods, but it is not clear how. Does LoopSeq bring anything new? And if yes, how. A more detailed description of how the method itself compares to previous approaches is needed.
2. Related to the point above, I didn't find the description of LoopSeq detailed enough. For instance how is the actual de-novo assembly of the barcoded sequences done? What is the fidelity of the assembly? What is its N50, and how accurately does it match the annotated isoforms? How many misassemblies were observed?
3. It's also not clear how LoopSeq compares to the previous methods in terms of accuracy? Are there any gains, and if there are, how would the authors justify them?
4. How do the observed expression values obtained by LoopSeq compare to previous methods? Is LoopSeq more, just as, or less accurate than other methods?
5. It is not clear how LoopSeq overcomes the known limitations of synthetic long read sequencing, as SLR methods usually generate low numbers of transcripts, most being incomplete at the 3' end. The authors should comment on this point, and explain why this is not the case with LoopSeq.
6. The fusion transcription method detection is not detailed enough. The authors mention two methods: one based on SQANTI and one based on text searches based on the 15bp ahead of and 15bp right after the junction point were applied to the long-read molecules. Why did the authors use both these methods? How do they compare? Was this latter method done manually for >2K isoforms? This doesn't seem plausible. Is 15 bp enough to do the searches? Why are 15bp enough as this size can not ensure uniqueness?

Minor comment:

1. Figure 2B is a little misleading as it overlaps isoforms and genes, and since multiple isoforms can belong to only one gene, the authors should clearly specify that this figure considers the genes associated with the isoforms, and not the isoforms themselves.

Reviewer #2 (Remarks to the Author):

The paper entitled “Targeted Transcriptome Analysis using Synthetic Long Read Sequencing Uncovers Isoform Reprogramming in the Progression of Colon Cancer” describes an RNA sequencing protocol LoopSeq and its applications to various transcriptome analysis. The authors apply their methods to sequencing ERCC synthetic RNA for validating both biotechnological and computational methods. In addition, as an example of possible application they perform three different types of analysis using LoopSeq data obtained from colon cancer samples: (i) quantification and differential isoform expression analysis, (ii) isoform-specific SNV calling and (iii) fusion gene discovery.

Long-read transcriptome sequencing enabled fundamentally new studies to be carried out. However, to my knowledge, no standard pipeline is yet established by the community. As PacBio and ONT platforms have issues of their own, both novel protocols and computational methods appear regularly.

LoopSeq appears to be a valuable alternative for analysing complex transcriptomes via sequencing of full-length isoforms and has a great potential for future research projects that involve studying genes at the isoform level. Moreover, as stated in the manuscript, LoopSeq is capable of generating far more accurate transcript sequences than long-read technologies.

Indeed, LoopSeq’s lower error-rate results in higher mapping quality (especially for splice junctions and short exons), and therefore increases accuracy of isoform detection (including novel and fusion transcripts).

Overall, the manuscript is well-structured and easy to follow. Below I summarize several questions and comments, which, in my opinion, may improve the quality of the current manuscript.

1. a) The authors provide almost no information on how short reads were assembled into full-length transcripts, which, in my opinion, is one of the most important steps of the analysis. The only mention is “LoopSeq employs a SLR sample preparation method that enables long read assembly, building upon previously published synthetic long read sequencing methodologies [10,11]”. To make LoopSeq data usable by other researchers, I suggest to clearly indicate assembly methods and make sure they are available and easy to run.

b) If the existing tool was used, I also recommend performing some benchmarks that

would show its performance specifically on LoopSeq data. Although sequencing ERCC is a solid validation of LoopSeq combined with the assembler, it may not provide insights on whether errors occur during sequencing or the assembly (see comments 3 and 4). Of course, the best benchmark would be simulating LoopSeq data with realistic properties (e.g. number of reads per UMI, transcript truncation distribution). For simulating Illumina short reads any available simulator can be used, e.g. InSilicoSeq [1], wgsim [2].

However, I do realise that this analysis may require significant effort and can be slightly out of scope of this work.

2. I suggest providing the exact commands and parameters used for all computational tools. Probably, a repository with a collection of scripts would be useful for reproducing the entire analysis, as well as future studies that will involve LoopSeq.

3. Figure 1D,E. In addition to existing figures, I recommend providing overall percentage of undetected transcripts, as well as their abundance compared to detected ones. Is it possible to verify, whether undetected molecules are caused by the assembly algorithm or a sequencing problem?

4. Although authors provide a substantial description of possible chimera formation mechanisms, a

similar question arises here: can some chimeras be formed during the assembly?

5. Regarding isoforms-specific SNVs, could some of them be the result of RNA editing?

6. Figure 1A. Notation "BC" is slightly confusing since in the text it is denoted as UMI.

7. Lines 102-104. The sentence looks a bit heavy to me.

8. I would suggest to put large supplementary tables (S1, S2, maybe some others) separately in the appropriate format (e.g. TSV), rather than making a thousand-page PDF.

9. Although choice of mimimap2 is reasonable, STARlong [3] and GMAP [4] may have slightly higher spliced alignment accuracy for long accurate sequences (such as LoopSeq contigs). They authors may consider using them as well.

References

1. Gourel, H., Karlsson-Lindsjö, O., Hayer, J. and Bongcam-Rudloff, E., 2019. Simulating Illumina metagenomic data with InSilicoSeq. *Bioinformatics*, 35(3), pp.521-522.

2. <https://github.com/lh3/wgsim>

3. Dobin, A., Davis, C.A., Schlesinger, F., Drenkow, J., Zaleski, C., Jha, S., Batut, P., Chaisson, M. and Gingeras, T.R., 2013. STAR: ultrafast universal RNA-seq aligner. *Bioinformatics*, 29(1), pp.15-21.

4. Wu, T.D. and Watanabe, C.K., 2005. GMAP: a genomic mapping and alignment program for mRNA and EST sequences. *Bioinformatics*, 21(9), pp.1859-1875.

Reviewer #3 (Remarks to the Author):

The manuscript by Liu et al investigates a modified approach to long read sequencing (LoopSeq) which is shown to generate long sequencing reads with enough depth to accurately define splicing isoforms in samples. The authors then applied this methodology to colon cancer samples and were able to show relevant splicing isoforms as well as novel gene fusion isoforms. Detailed look at the isoforms revealed that some SNVs switch between specific isoforms during cancer progression. Overall this manuscript is very interesting and does highlight the benefits of LoopSeq and its wider applicability.

Some of the concerns that need to be address before acceptance of the manuscript are:

- Figure 1 (B and C), it is not fully clear whether the contigs below or above the 0 mark (where most contigs are) are indeed cDNA molecules that are prematurely terminated during reverse transcription or rather real contigs with novel and previously unidentified TSS or TTS. Did the authors exhaust all way to test these explanations before concluding that these are simply technical errors?
- Figure 1 (G and H), the authors mention an overlap between LoopSeq error and Illumina error positions but no quantification is provided. It is left to the reader to gauge such level of overlap from a pretty low-resolution data figure.
- One of the main conclusions from this study is that pathological SNVs undergo isoform switching in cancer progression, yet none of the relevant data is in the main Figures, but rather in supplementary Figures and tables. If the authors are confident about this data, they need to move them to the main Figures.
- The panels in Figure 1 are not discussed in order in the text making it hard for the reader to follow the logic.

We thank the three reviewers for their constructive comments on the manuscript. We have made extensive revision based on these comments. Additional analyses were performed. New comparisons with the existing synthetic long-read sequencing methods were made. New details were added to the computational and sample preparation methodology. The revised manuscript represents a much-improved version of our work. Please see below point-by-point replies to the reviewer's comments:

Reviewer #1:

This manuscript describes a method that uses a barcoding-based synthetic long read sequencing to identify expressed isoforms from a transcriptome. The authors present it to be more accurate than alternative methods of identifying isoforms, and they apply it to samples from colon to identify isoform switching expression and specific SNVs in cancer progression. This is a noteworthy result, if it wasn't identified before. My main concerns with this manuscript are related to the description of the LoopSeq method and its accuracy, and are detailed below:

Comment #1. It is not clear how LoopSeq is different from previous methods. The authors cite a couple of methods that were previously developed, and they specify "building upon" these previous methods, but it is not clear how. Does LoopSeq brings anything new? And if yes, how. A more detailed description of how the method itself compares to previous approaches is needed.

Answer: We appreciate the reviewers comments over the distinguishing features of this work compared with previously published work. We have revised the manuscript in light of this comment, adding detail in the introduction, results, discussion and supplementary sections. We hope that the revisions are sufficient to explain the novel features of LoopSeq's SLR chemistry and software introduced to long-read transcriptome sequencing compared to state of the art.

Specifically, LoopSeq's unique and novel contribution to long-read transcriptome sequencing methodology that were added to the manuscript can be categorized as per the list below, with references to the manuscript sections and page numbers where the new descriptions and data were added:

1. Eliminating the need for physically compartmentalizing transcripts into many reaction chambers with different barcodes using microfluidics or 384 well plates (Introduction pages 1-2, Results section pages 2-3).
2. Introducing the ability to distinguish between highly homologous isoforms or highly similar edited transcripts that exist within the same sample (Introduction section page 4, Results section page 5)
3. Improving the ability to de-novo assemble SLR 3' and 5' termini by integrating synthetic adapters with known sequences at both termini.
4. Introducing the ability to determine whether SLRs contain the complete terminal 3' and 5' sequences of the full length transcripts by detecting synthetic adapters with known sequences at both termini, and reporting the assembly completeness status for each terminus of each long reads (Results section pages 5 & 6).

5. Reducing the per bp error rate of mRNA long-reads by increasing short read coverage uniformity along the sequence of barcoded transcripts/contigs (Results section pages 9-11 and supplementary figure S1).
6. Introducing q scores in fastq format for SLR's (Results section page 5, supplementary scripts and submitted fastq files).
7. Introducing methods for combining probe capture with SLR transcriptome sequencing (Results section page 11, Supplementary Methods pages 2-4).

Comment #2. Related to the point above, I didn't find the description of LoopSeq detailed enough. For instance how is the actual de-novo assembly of the barcoded sequences done? What is the fidelity of the assembly? What is its N50, and how accurately does it match the annotated isoforms? How many misassemblies were observed?

Answer: Thank you again for these comments. We added significant detail to the manuscript related to how the de novo assembly was performed, and the various metrics that we followed with regards to assembly fidelity and transcript annotation. Specifically, please see below a point by point reply to the questions raised in this comment with references to the manuscript pages where data and explanations were added:

- (1) Regarding how the actual de-novo assembly of the barcoded short reads to SLRs is performed, we included information in the Results section (page 4) about the assembler used and the parameters used to run it (Supplementary Scripts).
- (2) Regarding the fidelity of the assemblies, we included both a discussion and a quantitative comparative analysis to other long read and SLR methods over every source of error in the reads, including mismatches, indels and misassemblies based on the reference ERCC sequences (Results pages 9-11).
- (3) Regarding N50, we are not sure that N50 is optimal metric to look at for control samples since the exact expected sizes are known. N50 is also, in our opinion, not an optimal metric to follow for real transcriptome samples since variation in gene expression will change the N50, and therefore it is unclear what the "correct" N50 is and to benchmark results according to it. N50 is useful in examining read lengths in the context of genome sequencing, where larger N50 are always better, providing a simple framework for analysis. When comparing LoopSeq's reads to the expected lengths of the ERCC control RNA the lengths match up exactly.
- (4) With regards to mis-assemblies, the reviewer provides an excellent question that is key to the improvements in SLR technology. Mis-assemblies can arise either from the assembler itself or through the chemistry that can generate chimeric molecules. However, the contribution from chemistry versus the assembly cannot be decoupled. For example, chemistry issues during library prep can form chimeras between similar but non-identical transcripts, forming a hybrid molecules that are correctly assembled by the de novo assembler (using short reads from two different molecules that now share the same UMI), but represents a chemistry error. In other words, in these cases the de novo assembler perfectly assembles a sample preparation artifact. Likewise, an assembler may grab a bin of perfectly good short reads (that all originate from the same mRNA molecule) and

incorrectly assemble them due to, for example, repetitive elements that throw off the assembly. In both cases SLRs are made that do not fully map to the correct reference. However, the former example is a chemistry error while the latter is a de novo assembler error. With a known reference samples (ERCC standards in this case) we can use these so-called chimeras we find as a proxy to determine the total mis-assembly rate. We found this rate was 1.8% which means that at worst [Figure 1 Panel F], the mis-assembly rate (from chemistry and computational sources combined) in the LoopSeq method stands at that rate. Finally, our data show that the vast majority of mis-assemblies were formed between different molecules (i.e. had a 3' terminus of molecule A and 5' terminus of molecule B), not within molecules. Such molecule is a hallmark of PCR chimeras (example one above, not example two). Assembler generated mis-assemblies make contigs that are made entirely of sequences from the same molecule, not two different molecules (which is what we observed to be the vast majority of missassemblies). Nonetheless, we cannot rule out that some mis-assemblies do occur as a result of computational de novo assembler errors. Text describing this has been added to the Results section pages 8-9.

Comment #3 and #4. It's also not clear how LoopSeq compares to the previous methods in terms of accuracy? Are there any gains, and if there are, how would the authors justify them? How do the observed expression values obtained by LoopSeq compare to previous methods? Is LoopSeq more, just as, or less accurate than other methods?

Answer: This is an important comment as readers will be interested in and can certainly benefit from stacking up LoopSeq's accuracy against previously published methods. We revised the manuscript to include more comparative analysis and discussion. A Summary of our approach to the comparative analysis is detailed below (with references to manuscript pages), taking into account both traditional long reads and previously published SLR methods:

Our approach to benchmarking accuracy makes the assumption that the most informative way to compare LoopSeq to previous methods in terms of sequence accuracy and expression level accuracy is using the same reference sample sequenced with all long-read methods, in the case of mRNA the reference samples is a synthetic mix of known RNA molecules, a ground truth of samples (ERCC). Within this comparison it is critical to compare exactly the same metrics across all methods. PacBio and Oxford Nanopore previously published ERCC data, the same gold standard synthetic RNA mix used to evaluate LoopSeq in this paper. We provide a comparative analysis across all types of errors with commercially available sequencing technologies (PacBio and ONT). However, making a similar comparison to non-commercial, previously published SLR methods proved challenging due to the fact that previously published SLR technologies (Kuleshoc 2014, Zheng 2016, Stapleton 2016, Hong 2014) did not evaluate their error rates with ground truth RNA samples. Tilgner et al 2015 (which used an Illumina technology previously known as TruSeq SLRs, which has been abandoned/shelved a few years ago) is the only previously published SLR method that sequenced and published data from the ERCC reference sample, but focused most of its analysis on observed vs. expected expression, not on error rates, thereby making it difficult to directly compare their error rates to LoopSeq error rates. For

example, Tilgner et al did not report their mismatch rate, which is the largest source of sequencing error. Instead, they measured the fraction of long reads without indels (reported to be 4% of reads). Unfortunately, this does not inform us what the true per bp error rate is for the Tilgner et al method, since mismatches, the largest source of error, were excluded from their analysis. All other methods that sequenced the ERCC reference samples (PacBio, ONT and LoopSeq) did calculate the per bp error rate and it is reported in this manuscript (Sup Table S1). Additionally, Tilgner et al did not address the fact that the now shelved Truseq SLR's was not (by design) able to distinguish between highly homologous transcripts within the same sample, which is expected to increase the error rate in real samples, in which highly homologous isoforms are common, but not in the ERCC reference sample which does not contain highly homologous transcripts. Text describing the above has been added to the manuscript. It should also be mentioned that the Truseq SLR technology used in Tilgner et al (TruSeq SLR Library Prep method, Illumina, Cat No: 15047264) has been shelved/abandoned by Illumina roughly three years ago and is no longer available for further evaluation as a commercial product by the sequencing community. Text describing the above has been added to the results section of the manuscript (page 10).

ERCC expression levels are discussed (Results section, pages 7-8 and Figure 1E and 1D) and were found to be comparable in their accuracy to previously published ERCC expression data for PacBio, ONT and Tilgner et al.

Comment #5. It is not clear how LoopSeq overcomes the known limitations of synthetic long read sequencing, as SLR methods usually generate low numbers of transcripts, most being incomplete at the 3' end. The authors should comment on this point, and explain why this is not the case with LoopSeq.

Answer: Thank you for this comment. Explanation of Loopseq's transcript barcoding efficiency and its ability to more efficiently reconstruct 3' ends is now provided in the manuscript introduction section and the methods section. In short, Loopseq's barcode assignment step uses a barcoding primer that primes on the 3' poly-A tail of mRNA and employs template switching oligo opposite the 5' of the mRNA, similar to many non-SLR mRNA sequencing methods. Therefore, it generates comparable numbers of barcoded cDNA molecules compared to other methods for transcriptome sequencing that employs a similar mRNA to cDNA TSO-based conversion strategy. Regarding the sequencing of mRNA 3' termini, Loopseq has developed and integrated a new synthetic adapter at the 5' of the cDNA (3' of the mRNA) that enables accurate and efficient identification of the 3' end. Importantly, this synthetic adapter approach not only improves the ability to assemble the 3' of mRNA by trivializing its assembly with known, easy to assemble synthetic adapters, but it also trivializes the determination of whether the 3' ends have not been assembled properly (via 3' adapter identification) and reports that to the user, preventing false positive "discoveries" of novel 3' TTS's. Text describing this in more detail can be found in the Results section pages 5-6.

Comment #6. The fusion transcription method detection is not detailed enough. The authors mention two methods: one based on SQANTI and one based on text searches based on the 15bp ahead of and 15bp right after the junction point were applied to the long read molecules. Why

did the authors use both these methods? How do they compare? Was this latter method done manually for >2K isoforms? This doesn't seem plausible. Is 15 bp enough to do the searches? Why are 15bp enough as this size can not ensure uniqueness?

Answer: We appreciate the comments. We've updated the manuscript to clarify this in detail. Briefly, we have applied two methods to detect fusion transcripts/isoforms. One method is to apply SQANTI, which can annotate isoforms and discover fusion transcripts in an unbiased manner. Four novel fusions identified were also confirmed by minimap2 and STARlong aligner. The second method is to perform text searching on the known fusions selected from TCGA database. We first applied 15bp upstream and 15bp downstream around the junction point of a fusion to search for fusion isoforms as pre-filtering. For the high throughput searching on 20K+ fusions, we wrote a R script to search them automatically (see the supplementary files) instead of manually searching. For traditional short-read RNA-seq, 10bp or 15bp have been applied as minimum anchor length to search for fusion transcripts. So we just applied the same cutoff. We believe 4^{15} (around 1 billion) possibilities are powerful enough to serve as pre-filtering. Once fusion candidates were detected, we extended 15bp+15bp to the full long-read and applied NCBI BLAST tool to perform the alignment to confirm whether the candidate long-read was a true positive. For both fusion detection methods, SQANTI is a more unbiased discovery and text-search is a targeted identification. By applying both methods, we can discover fusion isoforms in a more comprehensive manner. Please refer to our updated result (subsection: Discovery of fusion gene isoforms) and method (subsection: Fusion Transcript detection) for details.

Minor comment:

Comment #7. Figure 2B is a little misleading as it overlaps isoforms and genes, and since multiple isoforms can belong to only one gene, the authors should clearly specify that this figure considers the genes associated with the isoforms, and not the isoforms themselves.

Answer: Thanks for the suggestion. In Figure 2, we performed the analysis at two levels: gene-level and isoform-level. At gene-level analysis, we quantified the read-count for whole gene expression, regardless which specific isoforms it belongs to. At isoform-level, we quantified read-count per isoform because LoopSeq long-read sequencing was able to reach isoform resolution. For example, if a gene can generate three isoforms, we will calculate read-count per isoform at isoform-level analysis, and sum up the read-count of all the three isoforms as the gene-level quantification. For Figure 2, we examined the overlapping between differentially expressed genes and isoform and then performed PCA and clustering respectively. The Venn diagram shows the count at gene-level for comparison. We've updated the manuscript (subsection: Quantification of gene and isoform expressions in human colon cancer) correspondingly.

Reviewer #2 (Remarks to the Author):

The paper entitled "Targeted Transcriptome Analysis using Synthetic Long Read Sequencing Uncovers Isoform Reprogramming in the Progression of Colon Cancer" describes an RNA sequencing protocol LoopSeq and its applications to various transcriptome analysis. The authors apply their methods to sequencing ERCC synthetic RNA for validating both biotechnological

and computational methods. In addition, as an example of possible application they perform three different types of analysis using LoopSeq data obtained from colon cancer samples: (i) quantification and differential isoform expression analysis, (ii) isoform-specific SNV calling and (iii) fusion gene discovery.

Long-read transcriptome sequencing enabled fundamentally new studies to be carried out. However, to my knowledge, no standard pipeline is yet established by the community. As PacBio and ONT platforms have issues of their own, both novel protocols and computational methods appear regularly.

LoopSeq appears to be a valuable alternative for analysing complex transcriptomes via sequencing of full-length isoforms and has a great potential for future research projects that involve studying genes at the isoform level. Moreover, as stated in the manuscript, LoopSeq is capable of generating far more accurate transcript sequences than long-read technologies. Indeed, LoopSeq's lower error-rate results in higher mapping quality (especially for splice junctions and short exons), and therefore increases accuracy of isoform detection (including novel and fusion transcripts).

Overall, the manuscript is well-structured and easy to follow. Below I summarize several questions and comments, which, in my opinion, may improve the quality of the current manuscript.

Comment #1a. The authors provide almost no information on how short reads were assembled into full-length transcripts, which, in my opinion, is one of the most important steps of the analysis. The only mention is “LoopSeq employs a SLR sample preparation method that enables long read assembly, building upon previously published synthetic long read sequencing methodologies [10,11]”. To make LoopSeq data usable by other researchers, I suggest to clearly indicate assembly methods and make sure they are available and easy to run.

Answer: We thank the reviewer for this comment. We think that adding information regarding how the short reads are assembled into SLR's has improved the quality of the manuscript. We have revised the manuscript by adding the information. Please see the introduction and supplementary scripts sections for details regarding how short reads were assembled into contigs and additional related tools used in the pipeline.

Comment #1b: If the existing tool was used, I also recommend performing some benchmarks that would show its performance specifically on LoopSeq data. Although sequencing ERCC is a solid validation of LoopSeq combined with the assembler, it may not provide insights on whether errors occur during sequencing or the assembly (see comments 3 and 4). Of course, the best benchmark would be simulating LoopSeq data with realistic properties (e.g. number of reads per UMI, transcript truncation distribution). For simulating Illumina short reads any available simulator can be used, e.g. InSilicoSeq [1], wgsim [2]. However, I do realise that this analysis may require significant effort and can be slightly out of scope of this work.

Answer: We agree with the reviewer that simulation work is outside the scope of this manuscript, especially in light of Yahav 2019 and Holzer 2019, that have extensively evaluated SPADES, the assembler used in this work. Nevertheless, we have provided additional data that

quantifies each type of error. Specifically, indel and mismatch errors are typically introduced by synthetic DNA and by DNA polymerases, respectively, not the assembler and both have been quantified in this manuscript. In the revised manuscript we include a new comparison of all types of errors found in the long reads against other previously published SLR and traditional long read approaches (Supplementary table S1). Perhaps most importantly, we found that the vast majority of “misassembled” reads were intermolecular chimeras (i.e. contain a 5’ from gene A and a 3’ from gene B) which suggests that the “misassembled” contigs are in fact chemistry (PCR chimera) related, not de novo assembly related. Having said that, we cannot rule out that there are other erroneous molecules that were generated by the assembler, but were misclassified by us as chemistry related. Text describing the above has been added to the manuscript introduction section.

Comment #2. I suggest providing the exact commands and parameters used for all computational tools. Probably, a repository with a collection of scripts would be useful for reproducing the entire analysis, as well as future studies that will involve LoopSeq.

Answer: We appreciate your suggestion. We’ve summarized all the bioinformatics pipelines applied for the LoopSeq long-read data in this paper, including read alignment, gene quantification, SNV calling and fusion identification. All the scripts are available at supplementary material.

Comment #3. Figure 1D,E. In addition to existing figures, I recommend providing overall percentage of undetected transcripts, as well as their abundance compared to detected ones. Is it possible to verify, whether undetected molecules are caused by the assembly algorithm or a sequencing problem?

Answer: The undetected/drop out transcripts are the transcripts marked Yellow at the lower end of the abundance spectrum of the ERCC expected abundance mix. Their expected abundance is the x axis. Based on the number of barcoded transcripts/cDNA molecules that were sequenced from the ERCC sample (roughly 66K long reads) and the expected abundance of these transcripts we expect these transcripts to drop out. An explanation of this has been added to the manuscript results section, the dropouts are marked with a different color in the figure and their expected concentration can be found on the x axis of the figure.

Comment #4. Although authors provide a substantial description of possible chimera formation mechanisms, a similar question arises here: can some chimeras be formed during the assembly?

Answer: With regards to chimera/mis-assemblies, this is an excellent observation that is central to improvements in SLR technology. Chimeras can arise either from the assembler itself (SPADES in this case) or by PCR, both of which will throw off the accuracy of an assembled SLR (relative to a known reference). For example, PCR can form chimeras between different transcripts, forming a hybrid molecule that can be assembled but represents a mistake in the process. Likewise, an assembler may grab a bin of short reads and incorrectly assemble them due to repetitive elements or mutations generated in sequencing that throws off the alignment. The contribution from chemistry versus an assembler cannot be easily decoupled. The chimera rate

has been analyzed in this manuscript (for ERCC), and while the analysis did not attempt to decouple PCR related chimeras from assembler errors, we believe that it did decouple them for two reasons. First, because we look for mismatched ends (e.g long-reads with a 5' terminus from an ERCC transcript A and a 3' terminus from a different transcript B). This type of error is exactly what a PCR chimera would look like, but not necessarily what an assembler misassembly would look like. Secondly and more importantly, all the chimeras/misassemblies that we found were between two completely different genes, suggesting that the misassembly was generated by a PCR chimera between two different molecules. An assembler misassembly would have short reads from the same gene, not two genes.

We used these misassemblies that we found as a proxy to determine the chimera rate. We found this rate was 1.8% which means that at worst [Figure 1 Panel F], the chimera rate in the LoopSeq method stands at that rate. Finally, it is worth mentioning that the rate of missassemblies that originate from the assembler will vary, to a large extent, with the sequence context of the transcript being assembled. A full evaluation of the misassembly rate of SPADES, the assembler used in this work, which is slightly out of scope of this work, has been previously published in the context of a side by side comparison with other de-novo assemblers in Yahav 2019 for genomes and Holzer 2019 for transcriptomes. Additionally, in the colon cancer samples, we identified fusion candidates based on the long-reads and in order to check whether these fusion isoforms are real chimeras or artificial ones, we have performed TaqMan RT-PCR and Sanger sequencing to validate these candidates. All the four novel fusions shown in Figure 5 were validated as true fusions using Sanger sequencing. Text describing the above concepts has been added to the manuscript (page 8). We thank the reviewer for this comment which helped make the manuscript better.

Comment #5. Regarding isoforms-specific SNVs, could some of them be the result of RNA editing?

Answer: We appreciate the reviewer's question. To avoid SNV called from transcriptome sequencing to be RNA-editing or sequencing error, we also performed whole exome sequencing (WES) on the same tissue to identify SNVs in DNA-level. The reported SNVs were detected from both RNA and DNA sequencing. Please refer to the result section for details (subsection: Expression pattern analyses of isoform-specific single nucleotide variants).

Comment #6. Figure 1A. Notation "BC" is slightly confusing since in the text it is denoted as UMI.

Answer: We thank the reviewer for identifying the inconsistency. We've changed 'BC' in Figure 1A into 'UMI' to keep the terms consistent.

Comment: #7. Lines 102-104. The sentence looks a bit heavy to me.

Answer: We thank the reviewer for the comment. We've modified this sentence accordingly.

Commnet #8. I would suggest to put large supplementary tables (S1, S2, maybe some others) separately in the appropriate format (e.g. TSV), rather than making a thousand-page PDF.

Answer: We thank the reviewer for pointing this out. We've formatted all the supplementary tables into Excel and Word format. Supplementary excel tables were individually uploaded. Methods, supplementary script, supplementary table/figure legends and supplementary figures were separately uploaded. This should make it easier for reader to access to the information.

Comment #9. Although choice of minimap2 is reasonable, STARlong [3] and GMAP [4] may have slightly higher spliced alignment accuracy for long accurate sequences (such as LoopSeq contigs). They authors may consider using them as well.

Answer: We appreciate that the reviewer refers these two long-read aligners to us. Originally, we tried two methods for fusion transcripts identification. The first method is to perform text searching on sequences around the known fusion junction point, which doesn't require read alignment but searches the raw read directly. The second method is to apply the tool SQANTI for long-read isoform annotation and fusion identification. This tool is actually based on GMAP aligner. For gene/isoform quantification, the read count is also based on SQANTI annotation. We applied minimap2 for long-read alignment and downstream SNV calling analysis. In the updated manuscript, we've also checked the four novel fusions from the minimap2 alignment and minimap2 is able to detect all the four novel fusions, either by long mismatch from the CIGAR info or by multiple/secondary alignment (soft and hard alignment by the CIGAR info). We've additionally tried STARlong alignment and three of the four fusions are confirmed by STARlong aligner by long mismatch CIGAR info. We've added these new pipelines into the corresponding result section (subsection: Discovery of fusion gene isoforms). In addition, we've summarized all the fusion detection details in Supplementary Table S13 and scripts for the long-read aligners are summarized in Supplementary script file.

Reviewer #3 (Remarks to the Author):

The manuscript by Liu et al investigates a modified approach to long read sequencing (LoopSeq) which is shown to generate long sequencing reads with enough depth to accurately define splicing isoforms in samples. The authors then applied this methodology to colon cancer samples and were able to show relevant splicing isoforms as well as novel gene fusion isoforms. Detailed look at the isoforms revealed that some SNVs switch between specific isoforms during cancer progression.

Overall this manuscript is very interesting and does highlight the benefits of LoopSeq and its wider applicability.

Some of the concerns that need to be address before acceptance of the manuscript are:

Comment #1. Figure 1 (B and C), it is not fully clear whether the contigs below or above the 0 mark (where most contigs are) or TTS. Did the authors exhaust all way to test these explanations before concluding that these are simply technical errors?

Answer: The reviewer has made a keen observation that we would like to discuss. Premature termination of the reverse transcription reaction at the 5' TSS is a valid possibility that is a challenge to distinguish unambiguously from novel start sites with all transcriptome sequencing methods that involve reverse transcribing RNA into cDNA, including LoopSeq, due to less than

optimal processivity of Reverse Transcriptases. Our method uses a temple-switching oligo (TSO) to capture the 5' TSS, which in theory should only attach to a capped mRNA (true full length). In practice it is possible that the TSO can still attach to a pre-terminated molecule, giving it the appearance of "full length" even though it is not, and thus not a novel start. Nearly all non-canonical TSSs identified were downstream -100+ bases, none in between, suggesting they might be the result of premature termination of the RT reaction. Nevertheless, we do not know for certain.

On the 3' TTS we use a poly-T oligo to initiate the RT reaction from the mRNA poly-A tail. As such, a true TTS should be well captured since that is the priming site. However, mis-priming may happen if the 3' end of a poly-T oligo overlaps past (upstream) the true poly-A start site (due to A-patches and T-patches acting permissive for exact pairing in terms of frame alignment) and thus skip over or before the last few bases when RT is initiated. To address this, we added two degenerate bases on the 3' of the poly-T oligo, which helps to align the oligo to the true poly-A start but may still cause skipping of the last few bases of the true TTS, which we see in our data. Random skipping explains the observed shifts that form a normal distribution around the correct, known TTS.

We added these interesting discussion points to the results section page 6 to reflect the response to this comment. We also added descriptions of 5' and 3' synthetic adapters that LoopSeq integrates at each transcript's terminus that improves our detection and assembly of transcript ends (pages 5-6).

Comment #2. Figure 1 (G and H), the authors mention an overlap between LoopSeq error and Illumina error positions but no quantification is provided. It is left to the reader to gauge such level of overlap from a pretty low-resolution data figure.

Answer: Thank you for this comment. Based on this comment we have added a data file that shows the mutation position (compared to the reference) for both the Illumina and LoopSeq reads. We also performed a pairwise correlation analysis to see whether mutations are shared between the datasets at a statistically significant level. A paragraph describing these results has been included in the Results section page 9 and the data file has been added as Supplementary Table S1. We also enlarge Figure 1G and 1H to make the graph clearer and more visible.

Comment #3. One of the main conclusions from this study is that pathological SNVs undergo isoform switching in cancer progression, yet none of the relevant data is in the main Figures, but rather in supplementary Figures and tables. If the authors are confident about this data, they need to move them to the main Figures.

Answer: Thank you for the suggestion. We've added SNV isoform analysis results to the new Figure 4.

Comment #4. The panels in Figure 1 are not discussed in order in the text making it hard for the reader to follow the logic.

Answer: We have rearranged the discussion to make the figure order in logical fashion.

REVIEWERS' COMMENTS:

Reviewer #1 (Remarks to the Author):

I am satisfied with the authors's answers to my concerns.

Reviewer #3 (Remarks to the Author):

As was my initial assessment of the manuscript by Liu et al, I still think this is a very interesting study with a method that has wide application potential. The few concerns I had were more than appropriately addressed by the authors and the necessary changes were made to the manuscript. As such I have no further objection to accepting the revised manuscript for publication.